# Core and Whole Body Vibration Exercise Influences Muscle Sensitivity and Posture during a Military Foot March

**DOI:** 10.3390/ijerph18094966

**Published:** 2021-05-07

**Authors:** Kaitlin D. Lyons, Aaron G. Parks, Oluwagbemiga Dadematthews, Nilophar Zandieh, Paige McHenry, Kenneth E. Games, Michael D. Goodlett, William Murrah, Jaimie Roper, JoEllen M. Sefton

**Affiliations:** 1School of Kinesiology, Auburn University, Auburn, AL 36849, USA; kaitlin.d.lyons.ctr@mail.mil (K.D.L.); aaron.g.parks.mil@mail.mil (A.G.P.); odd0003@auburn.edu (O.D.); nlz0005@auburn.edu (N.Z.); pmch02@gmail.com (P.M.); jroper@auburn.edu (J.R.); 2Department of Applied Medicine and Rehabilitation, Indiana State University, Terre Haute, IN 47809, USA; kenneth.games@indstate.edu; 3Department of Sports Medicine, Auburn University, Auburn, AL 36849, USA; goodlmd@auburn.edu; 4Educational Foundations, Leadership and Technology, Auburn University, Auburn, AL 36849, USA; wmm0017@auburn.edu

**Keywords:** military medicine, EMG, posture, visual analog scale, musculoskeletal injury

## Abstract

Military foot marches account for 17–22% of Army musculoskeletal injuries (MSI), with low back pain (LBP) being a common complaint. Core-exercise and whole-body vibration (WBV) have been shown to decrease LBP in patients with chronic low back MSI. This study investigated if WBV and/or core-exercise influenced LBP or posture associated with a military ruck march. A randomized control trial with three groups: (1) WBV and core-exercise (WBVEx); (2) core-exercise alone (Ex); and (3) control evaluated the effects of core-exercise and WBV on LBP during/after a two 8 K foot marches with a 35 lb rucksack. The intervention groups completed three weeks of core-exercise training with/without WBV. Outcome measurements included visual analog scale (VAS), algometer, posture and electromyography (EMG). LBP, pressure threshold, and posture were elevated throughout the foot march regardless of group. LBP remained elevated for 48 h post foot march (*p* = 0.044). WBVEx and Ex did not have a significant effect on LBP. WBVEx and Ex both decreased muscle sensitivity and increased trunk flexion (*p* < 0.001) during the second foot march (FM2). The 8 K foot marches significantly increased LBP. Core-exercise training with/without WBV decreases low back muscle sensitivity. WBV and core-exercise increases trunk flexion which may help improve performance and may influence LBP.

## 1. Introduction

Foot marches are ubiquitous within military units as they are vital to transporting mission essential equipment across the operational environment. During modern day warfare service members are required to routinely carry equipment exceeding 100 lbs [1]. Load carriage demands on today’s warfighter have led to increased non-combat musculoskeletal injuries (MSI) [1], and are the second leading cause of MSI in infantry units [2]. Service members carrying weights exceeding 30 pounds are 50–60 percent more likely to sustain an MSI [1].

Weight during load carriage is primarily carried on the service member’s back via a rucksack. Therefore, it is unsurprising that back MSI are among the most frequently observed foot march injury [1,3,4]. Researchers and military commands have investigated different equipment and carrying techniques in an effort to reduce MSI, including double-packs and different weight distribution arrangements [5,6,7,8]. Double packs distribute the load around the trunk in order to maintain a more normal center of mass which may decrease overall back pain [5]. However, these packs increased hip and neck pain and decreased the range of motion required to complete operational tasks [5].

Distribution of weight in the rucksack on the upper part of the back has been shown to reduce the forward lean of service members and thus decrease muscle strain and fatigue as compared to placing the weight low on the back [6]. However, placing the weight high on the back can be destabilizing when marching on uneven terrains [6]. Thus, placement of the weight high on the back may reduce risk of MSI when marching on even terrain, while placing the weight lower on the back may reduce MSI when marching on uneven terrain [7,8]. Weight distribution and rucksack design may help to mitigate some of the pain and discomfort service members experience during foot marching. However, the incidence of back MSI resulting from load carriage remains high. There is a clear need for more effective interventions to reduce MSI from load carriage.

A major cause of chronic low back pain (LBP) is weakness in abdominal musculature [9]. A variety of core muscle training programs have been used to successfully reduce chronic LBP [9]. More recently, whole body vibration (WBV) training has been used to reduce LBP and overall disability in patients with chronic LBP [10,11,12,13,14,15,16]. WBV is a low frequency, low amplitude, mechanical vibration exercise that has been shown to induce muscular contractions and influence neuromuscular potentiation through the tonic vibration reflex [17]. WBV with or without a combination of core exercises has been shown to increase core muscle activity and proprioception in patients with chronic LBP [10,11,12,13,14,15,16] and in healthy patients [18]. 

Core exercise training with or without WBV may strengthen core muscles and increase core muscle activation during foot marching to help maintain posture and reduce low back stress. The purpose of this study was to determine if core exercise training with or without WBV reduced LBP or influenced posture in healthy, active individuals during and after an eight-kilometer weighted military foot march. Given WBV’s ability to increase core muscle activation in healthy patients [18], the authors hypothesize that core exercise training with WBV will reduce low back pain greater then core exercise training alone.

## 2. Material and Methods

This study utilized a three by five repeated measures randomized experimental control design to determine the effects of a core exercise intervention with and without WBV on LBP and posture during and after a weighted eight-kilometer foot march. Participants were healthy, physically active adults. Interested participants completed a health questionnaire to determine eligibility and were consented by a member of the research team. Participants were required to be physically active between 18 and 35 years of age. Participants were excluded if they had any of the following conditions: acute inflammation or infection, acute joint disorders/arthroses, chronic migraine headaches, cardiovascular diseases, recent joint implants, metal or synthetic implants, gallstones, epilepsy, recent thrombosis or thrombotic complaints, current low back complaints, current or recent concussion, or pregnancy. This study was approved by the Auburn University Institutional Review Board (Protocol 19-211 MR 1907). This study has been registered as a clinical trial (ISRCTN12264516).

Participants were randomly divided into one of three groups: core exercise (Ex), WBV and core exercise (WBVEx), or the control group, using a random number generator by the lead investigator. All participants completed two, self-paced, eight-kilometer weighted foot marches separated by four weeks. Participants carried a 35-pound Modular Lightweight Load-Carrying Equipment (MOLLE) around a level indoor track during each foot march. Both foot marches were completed in the same direction around the indoor track. Each MOLLE was fitted and packed by an Active Duty Army Officer with over 15 years of experience in the Infantry to simulate military standard packing conditions. Following the first foot march (FM1) and a one-week recovery period, the Ex and WBVEx groups completed an intervention of core exercise training for a period of three weeks. Participants performed three sets of planks, side-planks, isometric squats, v-ups, bridges, and back extensions for 30 s each, three times a week (Figure 1). All core exercises, regardless of group, were completed on a side-alternating WBV platform (Galileo Med L Novotec Medical, Pforzheim, Germany) as illustrated in Figure 1. The WBV platform was turned on only during core exercises for the WBVEx group. The WBV frequency was set to 15 hertz with an amplitude of three millimeters (peak acceleration = 1.36 g) for isometric squatting and bridge exercises. This frequency with a low amplitude has been shown to increase muscle contraction of the erector spinae and rectus abdominis compared to lower frequencies in patients with chronic LBP [18]. The remaining exercises (planks, side-planks, v-up, and back extensions) were completed at a frequency of six hertz, due to the proximity of the participants head to the platform (peak acceleration = 0.21 g). All exercises were completed in bare feet. For safety participants were asked to lightly grip the WBV platform handrail during the isometric squat, but not support their weight. The control group completed their normal activity for four weeks between foot marches. The main variable of interest during this investigation was LBP. Secondarily, muscle soreness, muscle activation and posture were evaluated during the foot marches. 

### 2.1. Pain and Muscle Soreness Assessment

A 100-mm visual analog scale (VAS) was used to evaluate LBP pre-foot march, during (at four kilometers), post-foot march, one day and two days following each eight-kilometer foot march. The VAS has been shown to be a reliable measure of pain in the clinical setting [19] and has a minimal clinical important different (MCID) of 35 mm in patients with acute low back pain [20]. For each measurement participants were presented a new VAS ranging from “no pain” to “worst pain imaginable”. Participants were instructed to place a horizontal mark indicating their LBP along the scale at each time point. Additionally, an algometer (Force TenTM FDS Digital Force Gage, Wagner Instruments, Greenwich, CT, USA) was used to evaluate low back muscle soreness. The algometer has been shown to be a reliable method for muscle soreness of the trunk [21,22]. Bilateral marks for application of the algometer prior to the foot march were placed on each participant’s back, three centimeters lateral to the fourth vertebrae of the lumbar spine [23]. Algometer measurements were taken pre- and post-foot march and one and two days following the foot march. Participants were instructed to tell the researcher when they felt pain or discomfort from the pressure of the algometer. One practice trial was completed on each side of the back, followed by three alternating measurements. The average score for each time point was used for analysis.

### 2.2. Muscle Activation Assessment

Electromyography (EMG) was used to evaluate the effects of core exercise and WBV on muscle activation across the eight-kilometer foot march. EMG has been previously used to assess core trunk activation during gait [24]. EMG data were collected via four wireless Tringo Avanti Sensors (Delsys Inc., Natick, MA, USA). Surface EMG was selected as it has been shown to be a non-invasive, reliable measure of muscle activation [25]. Sensors were preset to a sampling rate of 2000 hertz for all data collections. Each area was shaved and abraded using a razor, alcohol and gauze pads and left exposed to dry for five minutes prior to placement of sensors. One researcher placed sensors on all participants to improve interrater reliability. EMG sensors were placed bilaterally on the rectus abdominis three centimeters lateral to the umbilicus, parallel to the muscle fibers [26,27]. Additionally EMG sensors were placed bilaterally on the erector spinae three centimeters lateral to first vertebra of the lumbar spinae, parallel to the muscle fibers [25,26,28].

Muscle activation was recorded during the first, middle, and last kilometer of the foot march. A Butterworth on board filter with a bandwidth of 20–450 hertz was used during collection. All raw EMG signals were high-pass filtered at 30 hz to reduce noise from cardiac muscle activity [29,30]. A root mean squared (125 ms) amplitude analysis normalized to the first kilometer of each foot march was then analyzed.

### 2.3. Posture Assessment

Posture was evaluated during the eight-kilometer foot march to evaluate the effects of core exercise and WBV on proprioception during a weighted military foot march. A Zephyr bioharness (Medtronics, Minneapolis, MN, USA) was placed across the participant’s chest with the Zephyr device located under the participants left arm. Posture was evaluated at the first, middle and last kilometer of the foot march. A posture of zero degrees indicated that the participant was standing in the vertical position, positive 90 degrees indicated that participant was in the prone position, and negative 90 degrees indicated that the participant was in the supine position. The zephyr bioharness device has been previously validated against a standard tilt table with a very strong relationship (r > 0.99) [31].

### 2.4. Statistical Analysis

Statistical analyses were completed using R statistical software [32] and R studio [33] with packages; dplyr [34], lme4 [35], reshape2 [36], emmeans [37], ggplot2 [38], psych [39]. A power analysis was completed prior to recruitment to determine sample size. Based on a large effect size measured in a previous low back pain study [13], power of 0.80, and a 0.05 alpha level, nine participants were required for each of the three groups. Mixed effects models were used to evaluate the effect of core exercise and WBV on LBP, pain pressure threshold of the low back, posture and muscle activation of the erector spinae and of the rectus abdominis. Fixed effects for the model included treatment group (Ex, WBVEx, Control), measurement time point (Pre, 4 km, Post, Day 1 Post, Day 2 Post), and foot march (FM1, FM2). Random effects for the model included the subject identification number, interaction of measurement time point and subject identification number, and the interaction of foot march and subject identification number. Post hoc analyses were used to test significant interactions and main effects. Additionally Cohen’s d effect sizes [40] were calculated for each variable. Assumptions of normality of residuals and homogeneity of residuals were evaluated using residuals of the mixed effects model. Assumptions of normality were violated for EMG muscle activation and the VAS. Both variables were log transformed to meet the assumption of normality. An a priori alpha level of 0.05 was used to determine significant results.

## 3. Results

Forty-one participants volunteered to complete this study. Two participants were dropped from the study; one was unable to complete all aspects of the study due a lack of time and one due to a foot injury (Figure 2). Thus, thirty-nine participants (female = 17, male = 22) completed all aspects of the study and were used for analysis. Demographics for the thirty-nine participants are in Table 1. No adverse or secondary effects were seen from the core exercise program with or without WBV.

### 3.1. Visual Analog Scale

There was no significant interaction between treatment group, foot march number (FM1 or FM2) and/or measurement time points for the VAS (Table 2). There was a main effect of foot march (F = 10.974, *p* = 0.002) and measurement time point (F = 70.796, *p* < 0.001) on the VAS, but no main effect of group (F = 0.444, *p* = 0.645). Regardless of foot march number (FM1, FM2), the VAS was significantly elevated four kilometers into the foot march (t = 4.638, *p* < 0.001), immediately following the foot march (t = 15.501, *p* < 0.001), one day (t = 4.899, *p* < 0.001), and two days following the foot march (t = 0.044, *p* = 0.044) as compared to prior to the foot march. VAS scores were also significantly elevated immediately following the foot march (t = 2.986, *p* = 0.003) as compared to the VAS scores at the four kilometer midpoint of the foot march and one day following the foot march (t = 10.640, *p* < 0.001), indicating that LBP continued to increase throughout the foot march, but decreased the following day. There was a significant difference in VAS scores between day one and day two following the foot march (t = −2.852, *p* = 0.004), indicating that LBP continued to decrease two days after the foot march. Lastly, VAS scores were significantly lower during FM2 as compared to FM1 (t = −2.701, *p* = 0.007) regardless of treatment group or measurement time point.

### 3.2. Algometer

One participant’s algometer measurements were removed prior to analysis because the participant refused to acknowledge the stimulus was painful until after the study (Figure 2). Readings between the two sides of the back were averaged for further analysis due to a strong correlation (r = 0.970) between left and right algometer readings for all participants. A significant interaction was indicated between foot march and treatment group for the algometer (F = 4.152, *p* = 0.024), however no other significant interactions were found. Post hoc analysis indicated a significant difference between algometer measurements between the WBVEx and control group during FM1(t = −2.290, *p* = 0.027) and FM2 (t = −3.791, *p* < 0.001). No difference was found between the WBVEx and Ex groups for FM1 (t = −1.625, *p* = 0.113) or FM2 (t = −1.987, *p* = 0.054). No difference was found between the Ex and control groups for FM1 (t = −0.619, *p* = 0.540) or FM2 (t = −1.729, *p* = 0.092).

There were main effects of measurement time point (F = 9.535, *p* < 0.001), foot march (F = 15.391, *p* < 0.001) and group (F = 4.856, *p* = 0.014). Regardless of foot march and group, algometer measurements were significantly decreased immediately following the foot march, (t = −3.725, *p* < 0.001) and one day following the foot march (t = −3.027, *p* = 0.003) as compared to prior to the foot march indicating a decrease in pressure pain threshold. However, there was no significant difference between algometer measurements two days following the foot march as compared to prior to the foot march (t = −0.394, *p* = 0.694). Additionally, there was no difference between immediately following the foot march and one day following the foot march (t = 0.698, *p* = 0.486), indicating that the pain pressure threshold remained constant for 24 h following the foot marches. Regardless of treatment group and measurement time point, algometer scores were significantly increased across time points for FM2 as compared to FM1, indicating an increase in pressure pain threshold as compared to the FM1 (t = 5.991, *p* < 0.001). The WBVEx group had a significantly increased algometer readings as compared to the control group (t = −3.097, *p* = 0.003) regardless of foot march or measurement time points, but no difference as compared to the Ex group (t = −1.842, *p* = 0.074). No difference was found between the Ex and control group (t = 1.192, *p* = 0.240). Effect sizes comparing FM1 and FM2 following the foot march are illustrated in Figure 3. 

### 3.3. Posture

A significant interaction was indicated between foot march and treatment group for posture (F = 3.635, *p* = 0.036), however no other interactions were found. Post hoc analysis indicated no significant difference in posture between groups during FM1. However, during FM2 the WBVEx group had a significant increase in trunk flexion posture as compared to the control group (t = −2.025, *p* = 0.049). The WBVEx also had an increase in trunk flexion posture during FM2 as compared to FM1 (t = 3.565, *p* < 0.001). The control group exhibited a decrease in trunk flexion posture during FM2 as compared to FM1 (t = −2.175, *p* = 0.031).

There was a main effect of measurement time point on posture (F = 76.238, *p* < 0.001), but no main effect was found for foot march (F = 0.471, *p* = 0.497) or group (F = 0.478, *p* = 0.478). Regardless of group and foot march, posture was significantly increased during the fourth kilometer (t = 5.932, *p* < 0.001) and the last kilometer (t = 8.090, *p* < 0.001) indicating an increase in trunk flexion compared to the first kilometer of the foot march. Additionally, there was a significant increase trunk flexion posture during the last kilometer as compared to the fourth kilometer (t = −2.180, *p* = 0.030).

### 3.4. EMG

A significant interaction was indicated between foot march and individual muscle for muscle activation (F = 3.563, *p* = 0.014), no other interactions were found. Post hoc analysis indicated that the left rectus abdominis was significantly more activated during FM1 and FM2 as compared to the right erector spinae (FM1: t = −2.592, *p* = 0.009, FM2: t = −2.865, *p* = 0.004) and left erector spinae (FM1: t = −2.145, *p* = 0.032, FM 2: t = −3.008, *p* = 0.002). The left rectus abdominis was not different than the right rectus abdominis during FM1 (t = 0.569, *p* = 0.570), however it was significantly increased during FM2 (t = −3.137, *p* = 0.002). Additionally, the left rectus abdominis had increased muscle activation during FM2 compared to FM1 (t = −2.551, *p* < 0.010). Similarly, the right rectus abdominis had an increase activation during FM1 as compared to the right erector spinae (t = −3.134, *p* < 0.002) and left erector spinae (t = −2.687, *p* < 0.007), however no difference was found during FM2. Additionally, there was no difference in activation during FM2 as compared to FM1 for the right rectus abdominis (t = −1.126, *p* = 0.260). The left erector spinae was not significantly different from the right erector spinae for FM1 (t = 1.876, *p* = 0.673) or FM2 (t = 0.155, *p* = 0.876). Additionally, the left erector spinae was not significantly different between foot marches (t = 1.876, *p* = 0.061), however the right erector spinae was significantly increased during FM2 (t = 2.323, *p* = 0.020).

There was a main effect of individual muscles on muscle activation (F = 2.978, *p* = 0.031). No main effect was found for foot march (F = 2.724, *p* = 0.108), measurement time point (F = 0.665, *p* = 0.515), or treatment group (F = 1.050, *p* = 0.357). Regardless of foot march, measurement time point and treatment group, the left rectus abdominis was significantly more activated then both the left (t = −3.586, *p* < 0.001) and right erector spinae (t = −3.806, *p* < 0.001). The right rectus abdominis had increased activation as compared to the right erector spinae (t = −1.980, *p* = 0.048). However, there was no difference between the left and right rectus abdominis (t = 1.772, *p* = 0.077) or left and right erector spinae (t = −0.195, *p* = 0.846).

## 4. Discussion

This project examined how a core exercise training program with and without WBV influenced posture, muscle activation, and LBP during and after an eight-kilometer weighted foot march. On average, participants experienced an increase in LBP throughout each foot march that remained elevated for two days when compared to prior to the foot march. LBP across the groups peaked immediately following the foot march and continued to decrease across the two follow-up days. Overall, LBP was significantly decreased during the second foot march as compared to the first foot march, regardless of group. Indicating that completing two foot marches within a month may decrease overall LBP during the second weighted eight-kilometer foot march. In novice participants, such as soldiers completing basic combat training, completing two identical foot marches prior to increasing load carriage weight may decrease LBP and low back MSI associated with load carriage. This recommendation is in line with the current literature showing that two foot marches a month increase foot march performance [41]. However, previous research did not evaluate if a reduction in LBP was a factor in the improved performance [41]. 

No statistical difference was found between groups for LBP as measured by the VAS. Core exercise training with/without WBV has been used as a means to treat participants with chronic LBP [9,10,11,12,13,14,15,42]. To the authors knowledge this is the first study that has used WBV to prevent or reduce LBP in otherwise healthy participants. It is possible that a lack of difference in LBP between groups may have been due to an overall low level of LBP generated in participants after the weighted foot march. All participants that completed the study were free from current LBP or injuries prior to the study. An increase of 38 mm and 27 mm on the VAS, indicating an increase in pain, were found immediately following FM1 and FM2, respectively. Approximately half of the participants during the first foot march did not have a clinically significant increase in LBP (MCID 35 mm [20]). Additionally, during the second foot march more than half of participants did not have a clinically significant increase in LBP. Thus, it is possible that the lack of induced LBP did not enable us to detect differences between the intervention groups. 

Pain pressure threshold measured with an algometer also decreased (indicating greater sensitivity) immediately following each foot march as compared to prior the foot march. However, contrary to the VAS results there was no difference in pain pressure threshold one or two days following the foot march. The VAS and algometry have previously been shown to be correlated [43], however we found no significant correlation between the two measurements. These two measurements assess a different component of back pain, and the VAS may be a better assessment of LBP at low levels. There was a medium effect of Ex group (*d* = 0.396) and WBVEx group (*d* = 0.592) on pain pressure threshold immediately following the foot march comparing FM1 to FM2, whereas only a small effect (*d* = −0.094) was found in the control group. This is a clinically relevant decrease in muscle sensitivity following the foot march which may enable service members to more effectively complete military tasks following a foot march.

An increase in overall trunk stiffness due to co-activation of the trunk flexors and extensors is typically seen during load carriage [44]. The results of our study indicate that, while both muscles were activated during the foot march, there was an increase in muscle activation of the rectus abdominis as compared to the erector spinae. These results are in line with previous research indicating an increase in rectus abdominis activation even in load carriage with light weight [45,46]. Load carriage of 10 percent of body weight has been shown to increase core muscle activation by 20–30 percent [45,46], and load carriage of 15 percent of body weight can increase core muscle activation by 54–105 percent [45,46]. Increases in core muscle activation result from the need to counterbalance the shift in center of mass resulting from the load on the back. The erector spinae have been shown to require a larger amount of load carriage weight before increases in muscle activation are seen. Load carriage weights from 63.9–103.6 pounds (29–47 kg) have been shown to increase erector spinae activation [47,48]. While carrying weights of less than 15 percent of body weight has been shown to decrease muscle activation of the erector spinae [47,48]. During the current study, participants carried a 35-pound (15.9 kg) rucksack, which was on average 21 percent of the participants’ bodyweight. This weight was chosen because it is the weight commonly used for entry level foot march training. Our participants were untrained, novice foot marchers and we did not want to put them at an increased risk for injury. The load carriage weight used may not have been heavy enough to provide a substantial increase in erector spinae activation (unloaded muscle activation patterns were not assessed during this study). Our results also revealed a significant increase in the left rectus abdominis activation as compared to the right rectus abdominis during FM2 across all groups. Previous literature has shown an increase in muscle activation of the right rectus abdominis as compared to the left [45,46]. In our study participants walked left around an indoor track which may have caused an increase in muscle activation on the left side as compared to the right. Bilateral differences in muscle activation patterns can increase the risk of MSI or pain.

The exercise training and/or WBV intervention did not increase rectus abdominis or erector spinae activation during FM2 as compared to the FM1. These results are contrary to previous literature showing that WBV or core muscle exercises increase core muscle activation in patients with chronic LBP. Patients with chronic LBP often have reduced core muscle activation which negatively effects their back pain levels [49]. The current study was completed by healthy participants free of chronic LBP to represent soldiers new to military training; therefore, they may not have had low core muscle activation prior to the intervention.

Participants in this study were not instructed on proper posture for a weighted military foot march. The WBVEx group had an increase in trunk flexion posture for FM2 as compared to FM1. It is possible that a combination of core exercise and WBV training strengthened the core musculature, allowing participants to safely increase forward flexion posture without increasing back pain. This is in line with previous research showing WBV has been effective in increasing proprioception in patients with chronic LBP [11,16]. Increases in trunk flexion have been associated with faster walking speeds [50]. This increase in trunk flexion posture would allow the participant to reposition their center of mass forward to increase forward momentum and possibly provide increases in speed or performance during the foot march. It is important to note that increase in forward flexion has been shown to increase the amount of compressive force on the low back as compared to a fully erect or supine position [51]. However, the increase in forward flexion for the WBVEx group did not increase LBP. There was actually a decrease the amount of LBP seen in the participants at four kilometers and at the end of the foot march. Anecdotally, participants noted that increases in flexion helped them walk at a faster pace during the foot march.

## 5. Limitations

The relatively light weight and short distance compared to weights and distances typically completed by experienced active duty service members may have reduced the amount of LBP seen during this study. The weight and distance chosen for this study were based on load carriage weights and distances used during initial entry soldier training and to reduce the chance of injury to our inexperienced participants. We also based the WBV treatment parameters on the previous literature for patients with chronic LBP. While low frequencies have been successful in reducing chronic LBP [10,11,12,13,14,42], higher frequencies have been used to induce greater muscle activation in healthy populations [52]. It is possible that frequency parameters and treatment length were insufficient to induce significant changes in the healthy population. We chose to complete this foot march on an indoor track to reduce the effect of environmental conditions (very hot southern summer weather) and outside confounding variables (traffic, distractions) that may have impacted this study. Foot marching on uneven terrain and inclines may have produced additional LBP. Lastly, since the lead investigator was responsible for supervising participants during the completion of their exercises, they were unable to be blinded to the randomization of groups throughout the study and analysis.

## 6. Conclusions

Two eight-kilometer weighted foot marches significantly increased LBP in novice participants. This LBP remained elevated for a minimum of two days following the eight-kilometer weighted foot march. Completing two foot marches within a month significantly decreased the amount of LBP that was seen in novice participants in the second foot march. A combination of core exercise and WBV training or core exercise training alone may have provided a clinically relevant decrease in muscle pain following the foot march. Additionally, a combination of core exercise and WBV training safely increased trunk flexion, which may have future implications on performance time. Lastly, we found an overall increase in rectus abdominis activation as compared to erector spinae activation. These findings may inform future recommendations on muscle strengthening for foot marching. Future research with core exercise and WBV training should be completed with higher weights, longer distances, and higher WBV frequencies. Additionally, future research may focus on service members that already are experiencing LBP or MSI as a result of military foot marches.

## Figures and Tables

**Figure 1 ijerph-18-04966-f001:**
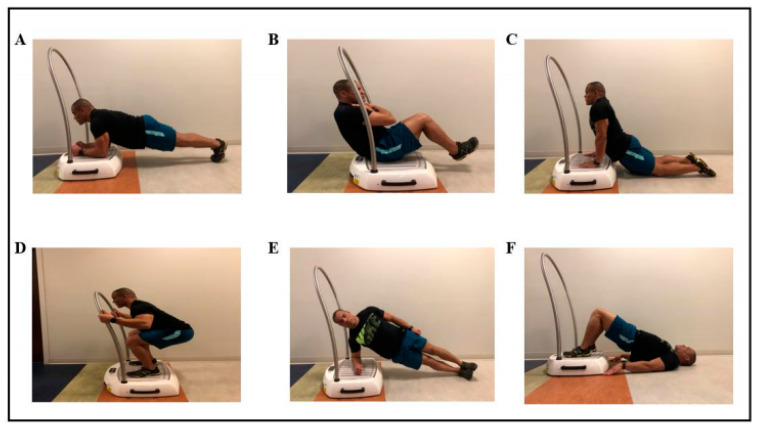
Core exercise training program. (**A**) Plank, (**B**) V-up, (**C**) Back extension, (**D**) Squat, (**E**) Side-plank, (**F**) Bridge.

**Figure 2 ijerph-18-04966-f002:**
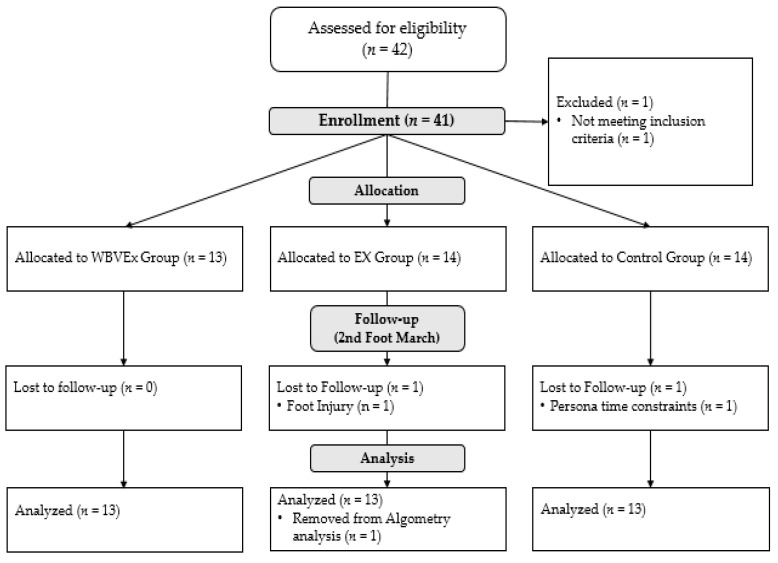
Flow diagram of study recruitment.

**Figure 3 ijerph-18-04966-f003:**
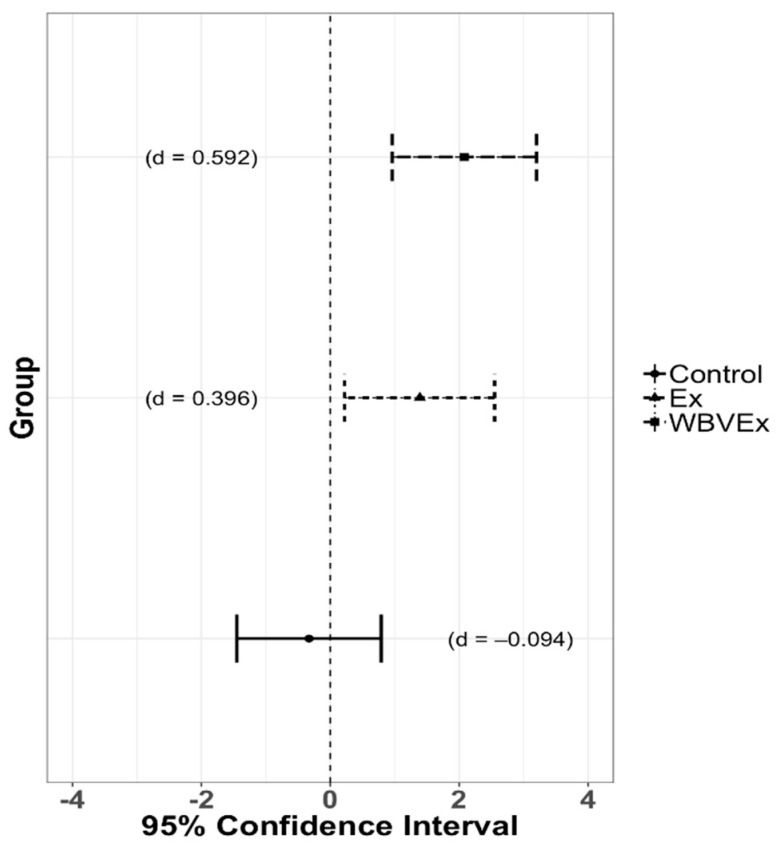
Algometer forest plot immediately following the foot march.

**Table 1 ijerph-18-04966-t001:** Summary of descriptive statistics.

Group	Age (yrs)	Height (cm)	Weight (kg)
Ex	22.8 ± 1.6	167.4 ± 9.2	69.9 ± 12.2
WBVEx	23.4 ± 3.9	173.8 ± 7.5	75.8 ± 12.2
Control	25.6 ± 5.4	172.4 ± 7.8	83.3 ± 15.1 *

Table 1 Legend: All data are presented as means ± standard deviation. Abbreviations: years old (yrs), centimeters (cm), kilograms (kg), whole body vibration and core exercise group (WBVEx), exercise group (Ex), * significant difference between weight of the Ex and control group (*p* < 0.05).

**Table 2 ijerph-18-04966-t002:** VAS, Algometer and Posture Before, During and After an 8 k Foot March.

		Foot March 1					Foot March 2				
		Pre	4 km	Post	1 Day Post	2 Day Post	Pre	4 km	Post	1 Day Post	2 Day Post
VAS (mm)	Ex	5.8 ± 10.3	38.3 ± 23.2	53.4 ± 30.3	18.9 ± 27.4	10.4 ± 18.5	3.8 ± 6.4	28.6 ± 20.8	40.7 ± 27.1	5.5 ± 8.5	5.5 ± 14.8
	WBVEx	4.0 ± 6.8	18.9 ± 14.3	31.2 ± 23.9	10.4 ± 17.0	4.7 ± 5.2	2.5 ± 4.1	16.6 ± 18.3	19.2 ± 22.4	6.5 ± 10.4	4.0 ± 5.7
	Control	2.2 ± 4.7	21.3 ± 16.4	42.4 ± 26.3	6.5 ± 15.2	6.5 ± 11.9	1.7 ± 3.4	15.6 ± 12.8	30.9 ± 21.0	5.8 ± 6.6	5.3 ± 5.8
Algometer (lbf)	Ex	7.8 ± 4.1	-	6.5 ± 2.7	6.4 ± 3.4	7.5 ± 3.4	8.5 ± 3.5	-	7.9 ± 3.6	8.6 ± 4.3	9.0 ± 4.4
	WBVEx	10.2 ± 4.8	-	8.2 ± 2.8	8.7 ± 2.7	9.8 ± 3.4	11.4 ± 4.5	-	10.3 ± 4.2	10.8 ± 4.4	11.9 ± 4.4
	Control	6.6 ± 2.7	-	6.3 ± 2.7	5.9 ± 2.9	6.2 ± 2.9	6.8 ± 2.9	-	5.9 ± 2.5	5.9 ± 2.7	6.6 ± 2.8
Posture (degrees)	Ex	17.7 ± 7.9	22.5 ± 7.3	24.9 ± 7.5	-	-	18.7 ± 6.2	23.2 ± 7.2	25.3 ± 6.1	-	-
	WBVEx	15.1 ± 5.7	19.4 ± 6.0	21.5 ± 6.3	-	-	19.9 ± 6.4	23.5 ± 6.1	24.9 ± 7.6	-	-
	Control	16.2 ± 6.1	22.9 ± 8.8	22.6 ± 8.5	-	-	16.1 ± 6.7	18.1 ± 6.1	19.9 ± 6.3	-	-

Table 2 Legend: All data are presented as means ± standard deviation. Abbreviations: millimeters (mm), pound-force (lbf), kilometer (km), whole body vibration and core exercise group (WBVEx), exercise group (Ex).

## Data Availability

The data presented in this study are available on request from the corresponding author.

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
