# Peer review of "Core and Whole Body Vibration Exercise Influences Muscle Sensitivity and Posture during a Military Foot March"

_ijerph, 2021, doi:10.3390/ijerph18094966_

Round 1

Reviewer 1 Report

The work is original and quite interesting, as it aims to reduce lower back pain in the military, a very common condition in the armed forces.
The work is well written, in terms of form and content. However, some very relevant information is missing to better understand how these measurements were made. That is, more information is needed on the characteristics of the two 8 km marches (whether they were the same route or not, in what time or at what speed they were executed, what percentages of slope they had, etc.). I suggest the authors to complete that information.
Lines 56-57: "More recently, whole body vibration (WBV) training has been used to reduce LBP and overall disability in patients with chronic": a reference is missing.

Author Response

Ms. Ana Stankovic and IJERPH Reviewers,

Thank you for reviewing our paper “Core and whole body vibration exercise influences muscle sensitivity and posture during a military foot march”. We appreciate all of your valuable feedback and have provide point by point feedback to each of your comments below. We appreciate your continued consideration of our manuscript for the International Journal of Environmental Research and Public Health.

Comments from Reviewer 1:

The work is original and quite interesting, as it aims to reduce lower back pain in the military, a very common condition in the armed forces.

The work is well written, in terms of form and content. However, some very relevant information is missing to better understand how these measurements were made. That is, more information is needed on the characteristics of the two 8 km marches (whether they were the same route or not, in what time or at what speed they were executed, what percentages of slope they had, etc.). I suggest the authors to complete that information.

Thank you for this comment. Information about the two 8km foot marches have been updated to reflect that they were completed on a level indoor track, were self-paced and were completed in the same direction both times (lines 86-90).

Lines 56-57: "More recently, whole body vibration (WBV) training has been used to reduce LBP and overall disability in patients with chronic": a reference is missing.

Thank you for this comment. We have added relevant citations to the above mentioned line within the manuscript (lines 56-57).

Reviewer 2 Report

Thank you very much for allowing me to review this article. I think it is an interesting topic and I congratulate the authors for the work done. Below you will find different comments with the idea of improving the article.

In material and methods section I have many doubts about the methodological quality (e.g., sample size, registration in clinicaltrials, I do not know if the CONSORT guidelines have been followed). Please find my comments below:

Abstract

- “Core-exercise training with/without WBV decreases low back muscle sensitivity. WBV and core-exercise increases trunk flexion which may help improve performance and may influence LBP”. I think it is fine for the authors to state that both therapies are effective but it would be nice to know which is MORE EFFECTIVE (between-group analysis and effect size to know it).

Introduction

I find the introduction very appropriate. I would only add a study hypothesis before or after the objective.

As an objective, I think it is also interesting to know which of the two interventions is more effective (differences between groups as well as within groups).

Material and Methods

- Is this study registered with clinicaltrials.gov? The CONSORT clinical trial guidelines require that all clinical trials be registered prior to the start of data collection.

- How was the sample size determined? It is necessary that it has been calculated and that it is explained in detail how it was done.

- It should be specified which is the main variable and which are the secondary variables.

- It should be explained how the randomization was performed, who performed it, which evaluators were blinded. Was the person who performed the intervention the same as the person who evaluated? Were they blinded, were they blinded, were they blinded?

- The reliability of the measuring instruments used and/or the minimal clinically relevant differences should be added (as far as possible).

Results

- It is necessary to add the flow chart as required by CONSORT.

- Were there any adverse and/or side effects in any patient? It is necessary to explain them one by one in case they exist. In case they do not exist, it is necessary to say that there were no adverse or secondary effects in any subject.

Discussion

I find it very adequate.

Conclusion

- I think it is fine for the authors to state that both therapies are effective but it would be nice to know which is MORE EFFECTIVE (between-group analysis and effect size to know it

Author Response

Comments from Reviewer 2:

Thank you very much for allowing me to review this article. I think it is an interesting topic and I congratulate the authors for the work done. Below you will find different comments with the idea of improving the article.

In material and methods section I have many doubts about the methodological quality (e.g., sample size, registration in clinicaltrials, I do not know if the CONSORT guidelines have been followed). Please find my comments below:

Thank you for you review and the feedback for improvements. This trial has been registered as a clinical trial and the trial number has been added to the paper (Lines 82-83). We have also gone through the CONSORT 2010 checklist to make sure our methodology has the required information.

ABSTRACT:

- “Core-exercise training with/without WBV decreases low back muscle sensitivity. WBV and core-exercise increases trunk flexion which may help improve performance and may influence LBP”. I think it is fine for the authors to state that both therapies are effective but it would be nice to know which is MORE EFFECTIVE (between-group analysis and effect size to know it).

Thank you for this comment. We did complete a between-groups analysis for each of our outcome variables. These are reported in our results sections as the main effect between groups. We did not find any significant differences between the two therapy groups for any of our outcome variables. Additionally, we did not find any large differences between effect sizes for the therapy groups. Given this we did not feel comfortable, based on the results, concluding that one therapy was significantly more effective than the other.

INTRODUCTION

- I find the introduction very appropriate. I would only add a study hypothesis before or after the objective.

- As an objective, I think it is also interesting to know which of the two interventions is more effective (differences between groups as well as within groups).

 Thank you for this comment. We have added our hypothesis that Core exercise training with WBV will reduce low back pain, greater then core exercise training alone (Line 67-69).

MATERIAL AND METHODS

- Is this study registered with clinicaltrials.gov? The CONSORT clinical trial guidelines require that all clinical trials be registered prior to the start of data collection.

Thank you for this comment. This study has been registered. The trial number (ISRCTN12264516) has been added to the method section (Lines 82-83).

- How was the sample size determined? It is necessary that it has been calculated and that it is explained in detail how it was done.

Thank you for this comment. A power analysis was completed prior to recruitment for this study to determine a minimum sample size. Information about this calculation has been added to the manuscript (Lines 162-164).

- It should be specified which is the main variable and which are the secondary variables.

 Thank you for this comment, the main variable of interest (LBP) and secondary variables (muscle soreness, muscle activation and posture) were added to the methods section (Lines 108-109).

- It should be explained how the randomization was performed, who performed it, which evaluators were blinded. Was the person who performed the intervention the same as the person who evaluated? Were they blinded, were they blinded, were they blinded?

Thank you for this comment. The lead investigator randomly assigned the participants using a random number generator (Lines 85-86). The lead researcher, who also completed the statistical analysis was not blind to the groups that the participants were participating in since they were supervising the participants during their core exercises to make sure they were properly completing all exercises. To indicate this with in the manuscript this has been acknowledged as a limitation (Lines 394-396). 

- The reliability of the measuring instruments used and/or the minimal clinically relevant differences should be added (as far as possible).

Reliability and validity of the outcome measurements have been added to the methods section. In addition, MCID for the VAS has been added (Lines 115-117, 121-122, 132-133, 156-158).

RESULTS

- It is necessary to add the flow chart as required by CONSORT.

 Thank you for this comment. The Consort flow chart has been added to the manuscript as figure 2.

- Were there any adverse and/or side effects in any patient? It is necessary to explain them one by one in case they exist. In case they do not exist, it is necessary to say that there were no adverse or secondary effects in any subject.

There were no adverse or secondary effects from the core exercise program with or without vibration. This was notes in the manuscript (Lines 187-188).

DISCUSSION

- I find it very adequate.

CONCLUSION

- I think it is fine for the authors to state that both therapies are effective but it would be nice to know which is MORE EFFECTIVE (between-group analysis and effect size to know it

Thank you for this comment. As mentioned above, based on our statistical analysis we were unable to conclude that one therapy was statistically greater than the other.

Reviewer 3 Report

Dear authors, 

Although the article could have a potential of interest, a series of elements that I consider essential for its consideration for publication are missing. I have not found references to the article's registry, neither in internet nor in the article itself. In addition, I have not found references on whether any guideline has been followed, nor on the inclusion or exclusion criteria, nor on how the sample size calculation has been carried out.

In my opinion, these data are essential for an article to be considered for publication. Furthermore, the results are difficult to understand due to a lack of data in the text or in tables. While the graphs give you an idea, it is the data that allows you to assess in detail what has happened. On the other hand, I have not been able to find Zephyx bioharness validation data on the company page (which is reference number 26).

Author Response

Comments from Reviewer 3:

Dear authors,

Although the article could have a potential of interest, a series of elements that I consider essential for its consideration for publication are missing. I have not found references to the article's registry, neither in internet nor in the article itself. In addition, I have not found references on whether any guideline has been followed, nor on the inclusion or exclusion criteria, nor on how the sample size calculation has been carried out.

In my opinion, these data are essential for an article to be considered for publication. Furthermore, the results are difficult to understand due to a lack of data in the text or in tables. While the graphs give you an idea, it is the data that allows you to assess in detail what has happened. On the other hand, I have not been able to find Zephyx bioharness validation data on the company page (which is reference number 26).

Thank you for reviewing our manuscript submission and providing feedback. The current study was previously registered as a clinical trial, “Effects of Whole Body Vibration and Core Exercise on Muscle Soreness and Performance during a Foot March” (Trial number ISRCTN12264516). This trial number has been added to the method section (Lines 82-83).

Inclusion and exclusion criteria for the study has been added to the methods section, along with a CONSORT flow chart (figure 2) to provide details of recruitment and participation in the study. Additionally, a sample size was calculated through a power analysis, prior to the recruitment phase of this study. This information has been updated in the method section (Lines 162-164).

We appreciate you feedback on the results section. We have removed two of the figures and added a table (table 2) that provides means and standard deviations for several variables of interest. We believe that this will provide more information to the readers.

The zephyr bioharness has been previously validated as a measure of posture.  The current citation (26) was the manual, which explains how to interpret posture in the system. We have removed this reference and added the reference on validation of the Zephyr bioharness for measuring posture.  We have added a reference that shows previous validation (Lines 156-158).

Thank you again for your reviews of our manuscript. Please let us know if we can provide you with any additional information to assist in the review of this manuscript.

Respectfully,

JoEllen Sefton

Round 2

Reviewer 2 Report

The authors have made changes that greatly improve the article. In my opinion this manuscript should be accepted. Congratulations.